# SURCO: LEARNING LINEAR SURROGATES FOR COMBINATORIAL NONLINEAR OPTIMIZATION PROBLEMS

## ABSTRACT

Optimization problems with expensive nonlinear cost functions and combinatorial constraints appear in many real-world applications, but remain challenging to solve efficiently. Existing combinatorial solvers like Mixed Integer Linear Programming can be fast in practice but cannot readily optimize nonlinear cost functions, while general nonlinear optimizers like gradient descent often do not handle complex combinatorial structures, may require many queries of the cost function, and are prone to local optima. To bridge this gap, we propose **_SurCo_** that learns linear Surrogate costs which can be used by existing Combinatorial solvers to output good solutions to the original nonlinear combinatorial optimization problem, combining the flexibility of gradient-based methods with the structure of linear combinatorial optimization. We learn these linear surrogates end-to-end with the nonlinear loss by differentiating through the linear surrogate solver. Three variants of `SurCo` are proposed: `SurCo-zero` operates on individual nonlinear problems, `SurCo-prior` trains a linear surrogate predictor on distributions of problems, and `SurCo-hybrid` uses a model trained offline to warm start online solving for `SurCo-zero`. We analyze our method theoretically and empirically, showing smooth convergence and improved performance. Experiments show that compared to state-of-the-art approaches and expert-designed heuristics, `SurCo` obtains lower cost solutions with comparable or faster solve time for two real-world industry-level applications: embedding table sharding and inverse photonic design.

## 1 INTRODUCTION

Combinatorial optimization problems with linear objective functions, like linear programming (LP) (Chvatal et al., 1983) and mixed integer linear programming (MILP) (Wolsey, 2007), have been extensively studied in operations research (OR). The resulting high-performance solvers like Gurobi (Gurobi Optimization, LLC, 2022) can solve industrial-scale optimization problems with ten of thousands of variables in a few minutes.

However, even with perfect solvers, one issue remains: the cost functions $f(\boldsymbol{x})$ in many practical problems are *nonlinear*, and the highly-optimized solvers mainly handle linear or convex formulations while real-world problems have less constrained objectives. For example, in embedding table sharding (Zha et al., 2022a) one needs to distribute embedding tables to multiple GPUs for the deployment of recommendation systems. Due to the batching behaviors within a single GPU and communication cost among different GPUs, the overall latency (cost function) in this application depends on interactions of multiple tables and thus can be highly nonlinear (Zha et al., 2022a).

To obtain useful solutions to the real-world problems, one may choose to directly optimize the nonlinear cost, which is either a black-box output of a simulator (Gosavi et al., 2015; Ye et al., 2019), or a cost estimator learned by machine learning techniques (e.g., deep models) from offline data (Steiner et al., 2021; Koziel et al., 2021; Wang et al., 2021b; Cozad et al., 2014). However, many of these direct optimization approaches either rely on human-defined heuristics (e.g., greedy (Korte & Hausmann, 1978; Reingold & Tarjan, 1981; Wolsey, 1982), local improvement (Voß et al., 2012; Li et al., 2021)), or resort to general nonlinear optimization techniques like gradient descent (Ruder, 2016), reinforcement learning (Mazyavkina et al., 2021), or evolutionary algorithms (Simon, 2013). While these approaches can work in practice, they may lead to a slow optimization process, in

particular when the cost function is expensive to evaluate, and they often ignore the combinatorial nature of most real-world applications (encoded in the feasible set $\boldsymbol{x} \in \Omega$).

In this work, we propose a systematic framework `SurCo` that leverages existing efficient combinatorial solvers to find solutions to nonlinear combinatorial optimization problems arising in real-world scenarios. There are three versions of SurCo, `SurCo-zero`, `SurCo-prior`, and `SurCo-hybrid`. In `SurCo-zero`, given a nonlinear *differentiable* cost $f(\boldsymbol{x})$ to be minimized, we optimize a *linear surrogate* cost $\hat{\boldsymbol{c}}$ so that the *surrogate optimizer* (SO) $\min_{\boldsymbol{x} \in \Omega} \hat{\boldsymbol{c}}^\top \boldsymbol{x}$ outputs a solution that is expected to be optimal w.r.t. the *original* nonlinear cost $f(\boldsymbol{x})$. Due to its linear nature, SO can be solved efficiently with existing solvers, and the surrogate cost $\hat{\boldsymbol{c}}$ can be optimized in an end-to-end manner by back-propagating *through* the solver (Pogančić et al., 2019; Niepert et al., 2021; Berthet et al., 2020). In `SurCo-prior`, we consider a family of nonlinear differentiable functions $f(\boldsymbol{x}; \boldsymbol{y})$, where $\boldsymbol{y}$ parameterizes problem descriptions. We train the linear surrogate $\hat{\boldsymbol{c}}(\boldsymbol{y})$ on a set of optimization problems (called the training set $\{\boldsymbol{y}_i\}$), and evaluate on a held-out problem $\boldsymbol{y}'$, by directly optimizing SO: $\boldsymbol{x}^*(\boldsymbol{y}') := \arg\min_{\boldsymbol{x} \in \Omega(\boldsymbol{y})} \hat{\boldsymbol{c}}^\top(\boldsymbol{y}')\boldsymbol{x}$, which avoids optimizing the cost $f(\boldsymbol{x}; \boldsymbol{y}')$ from scratch. Finally, in `SurCo-hybrid` we use initial surrogate costs predicted by a fully-trained `SurCo-prior` and then fine-tune the surrogate costs further using `SurCo-zero` to leverage both domain knowledge synthesized offline and information about the specific instance.

All versions of `SurCo` are evaluated in two real-world nonlinear optimization problems: embedding table sharding (Zha et al., 2022a), and photonic inverse design (Schubert et al., 2022). In both cases, we show that in the on-the-fly setting, `SurCo` achieves higher quality solutions in comparable or less runtime, faster optimization in wall-clock time with lower solution cost, thanks to the help of an efficient combinatorial solver; in `prior`, our method obtains better solutions in held-out problems compared to other methods that require training (e.g., reinforcement learning). More specifically, in table sharding `SurCo-zero` obtains between 14% to 85% improvement in solution quality with between 2% and 23% increase in runtime overhead compared to the greedy baseline, `SurCo-prior` obtains between 47% and 71% solution quality improvement against the state of the art RL-based table sharding algorithm Zha et al. (2022b). `SurCo-hybrid` obtains better solutions than either `SurCo-zero` or `SurCo-prior`, with a similar runtime overhead as `SurCo-zero`. In photonic inverse design, `SurCo-zero` finds 21% more viable solutions for the beam splitter and twice as many solutions for the wavelength demultiplexers with all problems solving successfully for the mode converter and bend problems, taking between 10% to 64% less time than the pass-through approach from previous work (Schubert et al., 2022). While the offline trained `SurCo-prior` misses some optimal solutions in the different settings, it frequently obtains solutions in 0.5% to 2% of the runtime due to not needing to evaluate the objective and perform gradient steps. Again, `SurCo-hybrid` is able to obtain solutions more often than the other approaches, with a runtime overhead comparable to `SurCo-zero`. We additionally present theoretical results that help motivate why training a model to predict surrogate linear coefficients exhibits better sample complexity than directly predicting the optimal solution (Li et al., 2018; Ban & Rudin, 2019).

## 2   PROBLEM SPECIFICATION

Our goal is to solve the following nonlinear optimization problem describe by $\boldsymbol{y}$:

$$\min_{\boldsymbol{x}} f(\boldsymbol{x}; \boldsymbol{y}) \qquad \text{s.t.} \quad \boldsymbol{x} \in \Omega(\boldsymbol{y}) \tag{1}$$

where $\boldsymbol{x} \in \mathbb{R}^n$ are the variables to be optimized, $f(\boldsymbol{x}; \boldsymbol{y})$ is the nonlinear differentiable cost function to be minimized, $\Omega(\boldsymbol{y})$ is the feasible region, typically specified by linear (in)equalities and integer constraints, and $\boldsymbol{y} \in Y$ are the problem instance parameters drawn from a distribution $\mathcal{D}$ over $Y$. For example, in the traveling salesman problem, $\boldsymbol{y}$ can be the distance matrix among cities. We often consider solving a family of optimization problems, described as $\boldsymbol{y} \in Y$.

**Differentiable cost function**. The nonlinear cost function $f(\boldsymbol{x}; \boldsymbol{y})$ can either be the result of a simulator made differentiable via finite differencing (e.g., JAX (Bradbury et al., 2018)), or a cost model that is learned from an offline dataset, often generated via sampling multiple feasible solutions within $\Omega(\boldsymbol{y})$, and recording their costs. The cost model often takes the form of a deep neural network. In this work, we assume the following property of $f(\boldsymbol{x}; \boldsymbol{y})$:

**Assumption 2.1** (Cost function). *During optimization, the cost function $f(\boldsymbol{x}; \boldsymbol{y})$ and its partial derivative $\partial f / \partial \boldsymbol{x}$ are accessible.*

Learning a good nonlinear cost model $f$ is highly non-trivial for practical applications (e.g., AlphaFold (Jumper et al., 2021), Density Functional Theory (Nagai et al., 2020), cost model for embedding tables (Zha et al., 2022a)) and is beyond the scope of this work.

**Evaluation Metric**. In real-world applications, querying $f$ can be slow and expensive, and thus a lower number of queries while getting better quality solution is the goal. We mainly focus on two aspects: how good the solution $\hat{x}$ is, by checking the value of $f(\hat{x}; y)$, and how many queries of the nonlinear function $f$ are needed during optimization in order to achieve the solution $\hat{x}$.

**Linear/nonlinear cost function**. When $f(x; y)$ is linear w.r.t $x$, and the feasible region can be encoded using mixed integer programs or other mathematical programs, the problem can be solved efficiently using existing scalable optimization solvers. When $f(x; y)$ is nonlinear, we propose `SurCo` that learns a surrogate linear objective function, which allow us to leverage these existing scalable optimization solvers, and which results in a solution that has high quality with respect to the original hard-to-encode objective function $f(x; y)$. We will elaborate in the following sections.

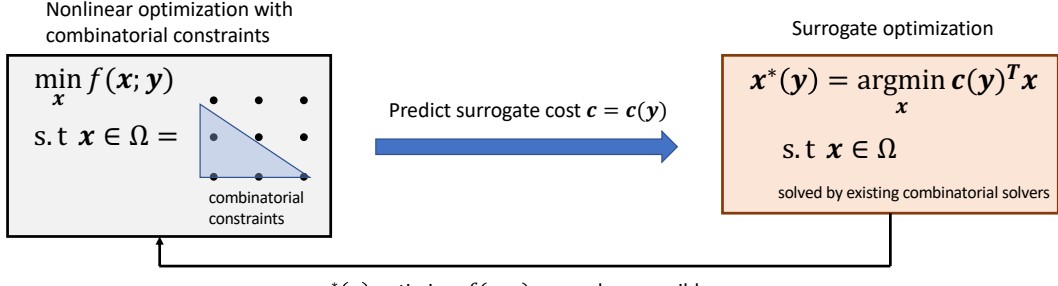

Figure 1: Overview of our proposed algorithm `SurCo`.

## 3 SURCO: LEARNING LINEAR SURROGATES

### 3.1 SURCO-ZERO: ON-THE-FLY OPTIMIZATION

We start from the simplest case in which we focus on a single instance with $f(x) = f(x; y)$ and $\Omega = \Omega(y)$. `SurCo-zero` aims to optimize the following objective:

$$(\texttt{SurCo-zero}): \quad \min_{c} \mathcal{L}_{\texttt{zero}}(c) := f(g_{\Omega}(c)) \tag{2}$$

where the surrogate optimizer $g_{\Omega} : \mathbb{R}^n \mapsto \mathbb{R}^n$ is the output of certain combinatorial solvers with linear cost weight $c \in \mathbb{R}^n$ and feasible region $\Omega \subseteq \mathbb{R}^n$. For example, $g_{\Omega}$ can be the following ($n$ is the number of variables to be optimized):

$$g_{\Omega}(c) := \arg\min_{x} c^{\top} x \quad \text{s.t.} \ \ x \in \Omega := \{Ax \leq b, x \in \mathbb{Z}^n\} \tag{3}$$

which is the output of a MILP solver. Thanks to previous works (Ferber et al., 2020; Pogančić et al., 2019), we can efficiently compute the partial derivative $\partial g_{\Omega}(c)/\partial c$. Intuitively, this means that $g_{\Omega}(c)$ can be *backpropagated* through.

Since $f$ is also differentiable with respect to the solution it is evaluating, we thus can optimize Eqn. 2 in an end-to-end manner using any gradient-based optimizer. That is, $c(t + 1) = c(t) - \alpha \frac{\partial g_{\Omega}}{\partial c} \frac{\partial f}{\partial x}$, where $\alpha$ is the learning rate. The procedure starts from a randomly initialized $c(0)$ and converges at a local optimal solution of $c$.

While Eqn. 2 is still nonlinear optimization and there is no guarantee about the quality of the final solution $c$, we argue that optimizing Eqn. 2 is better than optimizing the original nonlinear cost $\min_{x \in \Omega} f(x)$. Furthermore, while we cannot guarantee optimality, we are able to guarantee feasibility by leveraging a linear combinatorial solver. We note that `SurCo` is somewhat limited to problems without interior integer solutions, since the linear surrogate cannot yield interior points. However, many real-world settings, such as our two domains, consider making binary decisions which lack interior integer points. Intuitively, instead of optimizing directly over the solution space

$x$, we optimize over the space of surrogate costs $c$, and delegate the combinatorial feasibility requirements of the nonlinear problem to SoTA combinatorial solvers. Compared to naive approaches that directly optimize $f(x)$ via general optimization techniques, our method readily handles complex constraints of the feasible regions, and thus makes the optimization procedure easier. Furthermore, it also helps escape from local minima, thanks to the embedded search component of existing combinatorial solvers (e.g., branch-and-bound (Land & Doig, 2010) in MILP solvers). As we see in the experiments, this is particularly important when the problem becomes large-scale with more local optima. This approach works well when we are optimizing individual instances and may not have access to offline training data or the training time is cost-prohibitive.

### 3.2   SURCO-PRIOR: OFFLINE SURROGATE TRAINING

We now discuss more general cases, where the nonlinear loss function $f(x; y)$ represents *a family* of cost function to be optimized. Here the description of each *problem instance* $y$ is drawn from a fixed problem distribution $\mathcal{D}$. We then ask the following question: how can we find solutions to a batch of training instances $\mathcal{D}_{\text{train}} := \{y_i\}_{i=1}^N$, gain useful knowledge of the cost functions, and leverage such knowledge in held-out evaluation problem instances $\mathcal{D}_{\text{eval}}$ to accelerate the optimization procedure?

Following standard machine learning practice, let us first consider a naive two-stage approach. In the data collection stage, we simply apply SurCo-zero(Eqn. 2) to every $y_i$ separately to get $N$ surrogate cost vectors $c_i$. Then in the training stage, we train a regressor $\hat{c} = \hat{c}(y; \theta)$ on the dataset $\{(y_i, c_i)\}$ to learn to predict the surrogate costs from the problem features. Here $\hat{c}$ is a parameterized model (e.g., a deep network) with the parameters $\theta$ to be learned. This learned regressor $\hat{c}(y; \theta)$ can thus be used for a held-out problem instance $y'$ to directly predict $c' = \hat{c}(y'; \theta)$ and get the solution $x' = g_{\Omega(y')}(c')$ via surrogate optimizer (SO).

While this approach is simple, the $N$ optimization procedures in the data collection stage are independent of each other, and can lead to excessive number of calls to $f$ that are not helpful. E.g., if an optimization procedure converges to a bad local solution, then even if it achieves perfect convergence, which requires a lot of function calls, the resulting data point is still of low quality.

This motivates us to add a *regularizer* for the optimization:

$$(\text{SurCo-prior-}\lambda): \quad \min_{\theta, \{c_i\}} \mathcal{L}_{\text{prior}}(\theta, \{c_i\}; \lambda) := \sum_{i=1}^N f(g_{\Omega(y_i)}(c_i); y_i) + \lambda \|c_i - \hat{c}(y_i; \theta))\|_2 \quad (4)$$

Note that when $\lambda = 0$, it reduces to $N$ independent optimizations, while when $\lambda > 0$, the surrogate costs $\{c_i\}$ interact with each other. The intuition is that, the regressor $\hat{c}(y; \theta)$, even if not trained fully, can be very useful to guide $c_i$ rather than just using its randomly initialized version. Furthermore, if $\hat{c}$ is a mapping to global optimal solution of $x$, then it will pull the solutions out of local optima to re-target towards global ones, even when starting from poor initialization, yielding fast convergence and better final solutions for individual optimization instances.

A special case is when $\lambda \to +\infty$, we arrive at a novel objective that jointly learns the surrogate cost function, given the training set $\mathcal{D}_{\text{train}}$:

$$(\text{SurCo-prior}): \quad \min_{\theta} \mathcal{L}_{\text{prior}}(\theta) := \sum_{i=1}^N f(g_{\Omega(y_i)}(\hat{c}(y_i; \theta)); y_i) \quad (5)$$

This approach is useful when the goal is to find high-quality solutions for unseen instances of a problem distribution when the upfront cost of offline training is acceptable but the cost of optimizing on-the-fly is prohibitive. Here, we require access to a distribution of training optimization problems, but at test time only require the feasible region and not the nonlinear objective.

### 3.3   SURCO-HYBRID: FINE-TUNING A PREDICTED SURROGATE

Naturally, we consider SurCo-hybrid, a hybrid approach which initializes the coefficients of SurCo-zero with the coefficients predicted from SurCo-prior which was trained on offline data. This allows SurCo-hybrid to start out optimization from an initial prediction which has good performance for the distribution at large but which is then fine-tuned for the specific instance.

Formally, we initialize $c(0) = \hat{c}(y_i; \theta)$ and then continue optimizing $c$ based on the update from `SurCo-zero`. This approach is geared towards optimizing the nonlinear objective using a high-quality initial prediction that is based on the problem distribution and then fine-tuning the objective coefficients based on the specific problem instance at test time. Here, high performance comes at the runtime cost of both having to train offline on a problem distribution as well as performing fine-tuning steps on-the-fly. However, this additional cost is often worthwhile when the main goal is to find the best possible solutions by leveraging synthesized domain knowledge in combination with individual problem instances as arises in chip design (Mirhoseini et al., 2021) and compiler optimization (Zhou et al., 2020).

## 3.4 COST REGRESSION VERSUS SOLUTION REGRESSION: A THEORETICAL ANALYSIS

We also want to compare `SurCo` with the previous works on ML optimizers (Ban & Rudin, 2019) that try to directly learn the mapping from problem description $y$ to the solution, i.e. solution regression. Given a set of training instances $\mathcal{D}_{\text{train}}$ from distribution $\mathcal{D}$, these approaches first collect a set of training samples $\mathcal{D}_{\text{direct}} := \{y, x^*(y) : y \in \mathcal{D}_{\text{train}}\}$, and then learn a function $\tilde{x}^*(y)$ to fit the training samples.

While this is conceptually simple, there exist fundamental difficulties to learn such a direct mapping. First, as mentioned above, it can be quite expensive to obtain the optimal solution $x^*(y)$ due to the nature of nonlinear optimization and the query cost. Second, even if a perfect dataset $\mathcal{D}_{\text{direct}}$ is accessible, the number of samples needed to learn a mapping to directly predict $x^*(y)$ is related to the *Lipschitz constant $L$* of the mapping, and for a direct mapping, $L$ can be very large.

### 3.4.1 LIPSCHITZ CONSTANT AND SAMPLE COMPLEXITY

Let us first consider the sample complexity of solution regression methods as described above.

Formally, consider fitting any function $\phi : \mathbb{R}^d \supseteq Y \mapsto \mathbb{R}^m$ with a dataset $\{y_i, \phi_i\}$. Here $Y$ is a compact region with finite volume $\text{vol}(Y)$. The Lipschitz constant $L$ is the smallest number so that $\|\phi(y_1) - \phi(y_2)\|_2 \leq L\|y_1 - y_2\|_2$ holds for any $y_1, y_2 \in Y$. The following theorem shows that if the dataset covers the space $Y$, we could achieve high accuracy prediction: $\|\phi(y) - \hat{\phi}(y)\|_2 \leq \epsilon$ for any $y \in Y$.

**Definition 3.1** ($\delta$-cover). *A dataset $\mathcal{D}_{\text{direct}} := \{(y_i, \phi_i)\}_{i=1}^N$ $\delta$-covers the space $Y$, if for any $y \in Y$, there exists at least one $y_i$ so that $\|y - y_i\|_2 \leq \delta$.*

**Lemma 3.1** (Sufficient condition of prediction with $\epsilon$-accuracy). *If the dataset $\mathcal{D}_{\text{direct}}$ $(\epsilon/L)$-cover $Y$, then for any $y \in Y$, a 1-nearest-neighbor regressor $\hat{\phi}$ leads to $\|\hat{\phi}(y) - \phi(y)\|_2 \leq \epsilon$.*

**Lemma 3.2** (Lower bound of sample complexity for $\epsilon/L$-cover). *To achieve $\epsilon/L$-cover of $Y$, the size of the training set $N \geq N_0(\epsilon) := \frac{\text{vol}(Y)}{\text{vol}_0} \left(\frac{L}{\epsilon}\right)^d$, where $\text{vol}_0$ is the volume of unit ball in $d$-dimension.*

Please find all proofs in the Appendix. While we do not rule out a more advanced regressor than 1-nearest-neighbor that leads to better sample complexity, the lemmas demonstrate that the Lipschitz constant $L$ plays an important role in sample complexity.

### 3.4.2 DIFFERENCE BETWEEN COST AND SOLUTION REGRESSION

In the following we will show that in certain cases, the direct prediction $y \mapsto x^*(y)$ could have an infinitely large Lipschitz constant $L$.

To show this, let us consider a general mapping $\phi : \mathbb{R}^d \supseteq Y \mapsto \mathbb{R}^m$. Let $\phi(Y)$ be the image of $Y$ under mapping $\phi$ and $\kappa(Y)$ be the number of connected components for region $Y$.

**Theorem 3.1** (A case of infinite Lipschitz constant). *If the minimal distance $d_{\min}$ for different connected components of $\phi(Y)$ is strictly positive, and $\kappa(\phi(Y)) > \kappa(Y)$, then the Lipschitz constant of the mapping $\phi$ is infinite.*

Note that this theorem applies to a wide variety of combinatorial optimization problems. For example, when $Y$ is a connected region and the optimization problem can be formulated as an integer program, the optimal solution set $x^*(Y) := \{x^*(y) : y \in Y\}$ is a discrete set of integral vertices,

so the theorem applies. Combined with analysis in Sec. 3.4.1, we know the mapping $\boldsymbol{y} \mapsto \boldsymbol{x}^*(\boldsymbol{y})$ is hard to learn even with a lot of samples.

On the other hand, the mapping $\boldsymbol{y} \mapsto \boldsymbol{c}(\boldsymbol{y})$ can avoid too many connected components in its image $\boldsymbol{c}(Y)$, by connecting disjoint components of $\boldsymbol{x}^*(Y)$ together.

## 4 EMPIRICAL EVALUATION

We evaluate the two variants of `SurCo` on two real-world settings, embedding table sharding and inverse photonic design. Both have industrial application. Each setting consists of a family of problem instances with varying feasible region and nonlinear objective function.

### 4.1 EMBEDDING TABLE SHARDING

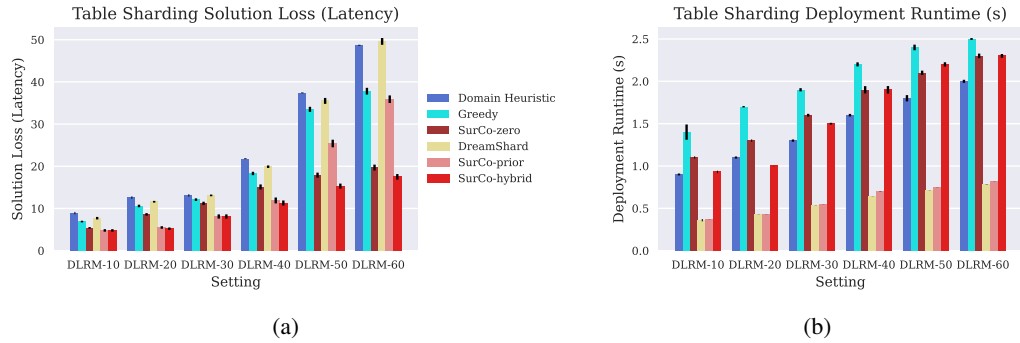

(a)                                                    (b)

Figure 2: Table placement plan latency **(a)** and solver runtime **(b)**. We evaluate `SurCo` against Dreamshard (Zha et al., 2022b) a SoTA offline RL sharding tool, a domain-heuristic of assigning tables based on dimension, and a greedy heuristic based on the estimated runtime increase. Striped approaches require pre-training.

The task of sharding embedding tables arises in the deployment of large scale neural network models which operate over both sparse and dense inputs (e.g., in recommendation systems (Zha et al., 2022a;b; Sethi et al., 2022)). Given $T$ embedding tables and $D$ homogeneous devices, the goal is to distribute the tables among the devices such that no device's memory limit is exceeded, while the tables are processed efficiently. Formally, let $x_{t,d}$ be the binary variable indicating whether table $t$ is assigned to device $d$, and $\boldsymbol{x} := \{x_{t,d}\} \in \{0,1\}^{TD}$ be the collection of the variables. The optimization problem is:

$$\min_{\boldsymbol{x}} f(\boldsymbol{x}; \boldsymbol{y}) \quad \text{s.t.} \quad \boldsymbol{x} \in \Omega(\boldsymbol{y}) := \left\{ \boldsymbol{x} : \quad \forall t, \sum_t x_{t,d} = 1, \quad \forall d, \sum_t m_t x_{t,d} \leq M \right\} \quad (6)$$

Here the problem description $\boldsymbol{y}$ includes table memory usage $\{m_t\}$, and capacity $M$ of each device. $\sum_d x_{t,d} = 1$ means each table $t$ should be assigned to exactly one device, and $\sum_t m_t x_{t,d} \leq M$ means the memory consumption at each device $d$ should not exceed its capacity. The nonlinear cost function $f(\boldsymbol{x}; \boldsymbol{y})$ is the *latency*, i.e., the runtime of the longest-running device. Due to shared computation (e.g., batching) among the group of assigned tables, and communication costs across devices, the objective is highly nonlinear. $f(\boldsymbol{x}; \boldsymbol{y})$ is well-approximated by a sharding plan runtime estimator proposed by Dreamshard (Zha et al., 2022b).

`SurCo` learns to predict $T \times D$ surrogate cost $\hat{c}_{t,d}$, one for each potential table-device assignment. During training, the gradients through combinatorial solver $\partial \boldsymbol{g}/\partial \boldsymbol{c}$ are computed via CVXPYLayers (Agrawal et al., 2019a) and the integrality constraints are relaxed. We found that in practice, we obtained solutions that were mostly integral in that only one table on any given device was fractional. At test time we solve for the integer solution using SCIP (Achterberg, 2009).

**Settings.** We evaluate `SurCo` on the publicly available Deep Learning Recommendation Model (DLRM) dataset (Naumov et al., 2019). We consider 6 settings: 10, 20, 30, 40, 50, and 60 tables are placed to 4 devices with each GPU device having a 5GB memory limit. Each setting has 100 problem instances (50 training and 50 test).

**Baselines.** For `SurCo-zero` baselines, we use `Greedy` that greedily allocates tables to devices while observing memory limits according to the predicted latency $f$, and `Domain-Heuristic`, the domain-expert algorithm of allocating tables to balance the aggregate dimension (Zha et al., 2022b). For `SurCo-prior`, we use Dreamshard, the SoTA embedding table sharding algorithm that requires training an offline RL policy.

**Results.** Fig. 2, `SurCo-zero` finds lower latency sharding plans than the baselines, while it takes slightly longer than `Domain-Heuristic` and DreamShard due to taking optimization steps rather than selecting based on a heuristic feature or reinforcement learned policy. `SurCo-prior` obtains lower latency solutions in about the same time as DreamShard with a slight increase in overhead due to using SCIP (Achterberg, 2009), a branch and bound MILP solver. Lastly, `SurCo-hybrid` obtains the best solutions in terms of solution quality and has runtime comparable to `SurCo-zero` since at test time it performs similar operations. In smaller problem instances ($T = 10$ to $T = 40$), `SurCo-prior` obtains better quality solutions than its impromptu counterpart, `SurCo-zero`, likely due to training on a variety of examples and being able to better escape local optima in any given problem instance as might be the case with the impromptu solver. However, as the problem size increases and more tables are available for placement, `SurCo-zero` gives better performance by optimizing for the test instances in question as opposed to `SurCo-prior` which only uses training data to obtain surrogate costs. Using `SurCo-hybrid`, we are able to obtain the best quality solutions but incur the upfront cost of pretraining and the deployment-time cost of optimizing the coefficients on-the-fly.

## 4.2 INVERSE PHOTONIC DESIGN

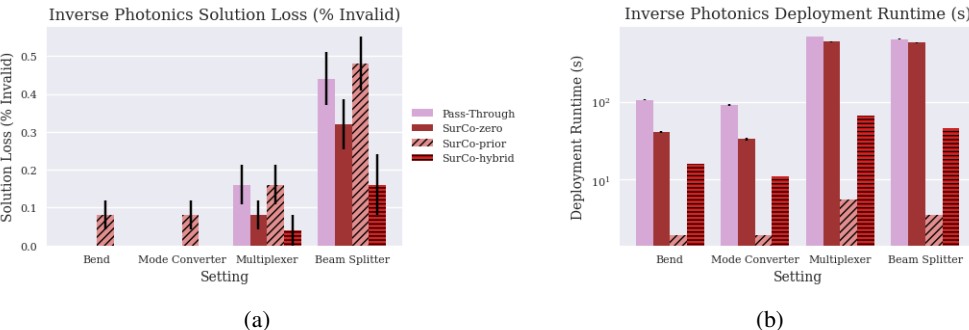

(a)  (b)

Figure 3: **(a)** The solution loss (% of failed instances when the design loss is not 0), and **(b)** test time solver runtime in log scale. For both, lower is better. We compare against the pass-through gradient approach proposed in Schubert et al. (2022). We observe that `SurCo-prior` achieves similar success rates to the previous approach `Pass-through` with a substantially improved runtime. Additionally, `SurCo-zero` runs comparably or faster, while finding more valid solutions than `Pass-through`. `SurCo-hybrid` obtains valid solutions most often and is faster than `SurCo-zero` at the expense of pretraining. Striped approaches use pretraining.

Photonic devices play an important role in high-speed communication (Marpaung et al., 2019), quantum computing (Arrazola et al., 2021), and machine learning hardware acceleration (Wetzstein et al., 2020). The photonic components can be formulated as a binary 2D grid, with each cell being filled or void. There are constraints for binary patterns: only those that can be drawn by a physical brush instrument with certain cross shape can be manufactured.

It remains challenging to find designs that are manufacturable and satisfy design specifications (e.g. beam splitting). An example solution developed by `SurCo` is shown in Figure 4b: coming from the top, beams are routed to the left or right, depending on wavelength. The solution is also manufacturable: a 3-by-3 brush cross can fit in all filled and void space.

Given the design, existing work (Hughes et al., 2019) enables differentiation of the design misspecification cost, evaluated as how far off the transmission of the wavelengths of interest is from the desired locations, with zero design loss meaning that the specification is satisfied. Researchers also develop a standard benchmark of inverse photonic design problems (Schubert et al., 2022).

**Settings.** We compare our approaches against the "Pass-Through" method (Schubert et al., 2022) on randomly generated instances of the four types of problems in Schubert et al. (2022): Waveguide Bend, Mode Converter, Wavelengths Division Multiplexer, and Beam Splitter. We generate 50 instances in each setting (25 training/25 test). Further generation details are in the appendix. We evaluated several algorithms described in the appendix, such as genetic algorithms and derivative-free optimization, which failed to find physically feasible solutions. We consider two wavelengths (1270nm/1290nm), and optimize at a resolution of 40nm, visualizing the test results in Fig. 3.

**Results.** Fig. 3, `SurCo-zero` consistently finds as many or more valid devices compared to the `Pass-Through` baseline (Schubert et al., 2022). Additionally, since the on-the-fly solvers stop when they either find a valid solution, or reach a maximum of 200 steps, the runtime of `SurCo-zero` is slightly lower than the `Pass-Through` baseline. `SurCo-prior` obtains similar success rates as `Pass-Through` while taking two orders of magnitude less time as it does not require expensive impromptu optimization, making `SurCo-prior` a promising approach for large-scale settings or when solving many slightly-varied instances. Lastly, `SurCo-hybrid` performs best in terms of solution loss, finding valid solutions more often than the other approaches. It also takes less runtime than the other on-the-fly approaches since it is able to reach valid solutions faster, although it still requires optimization on-the-fly so it takes longer than `SurCo-prior`. We visualize the convergence of impromptu solvers in Fig. 4a where `SurCo-zero` has smoother and faster convergence compared to the `Pass-through` approach.

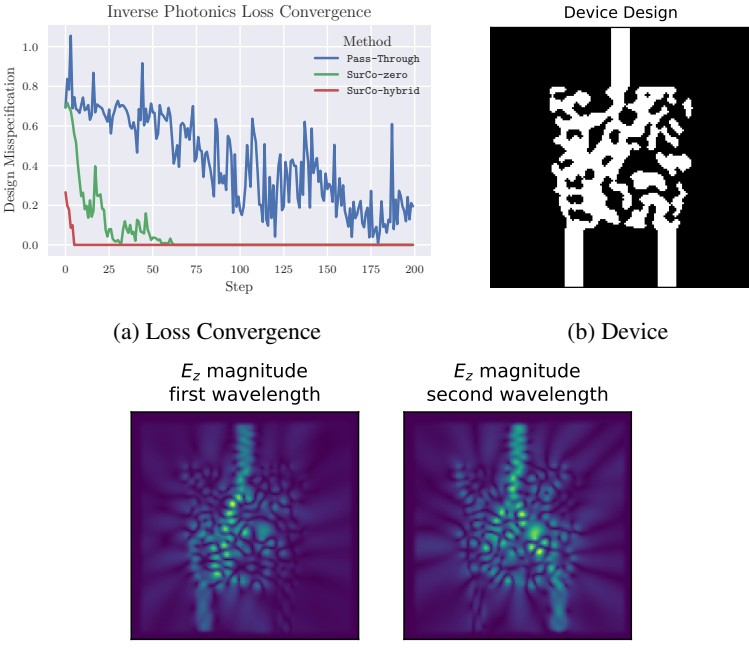

(a) Loss Convergence           (b) Device

(c) Wave Mangitude

Figure 4: Inverse photonic design convergence for a single instance (Schubert et al., 2022). `SurCo-zero` smoothly lowers the loss while the pass-through baseline converges noisily. Also, `SurCo-hybrid` starts out with a high-quality solution and fine-tunes until an optimal solution is reached. We also visualize the `SurCo-zero` solution with magnitudes of the two wavelengths of interest which we successfully route from the input at the top to the two different waveguides at the bottom.

## 5   RELATED WORK

**Differentiable Optimization**    Previous work differentiated through several optimization problems, calculating how changes in input parameters impact the optimal solution. Initially, a differentiable convex quadratic programming solver called OptNet (Amos & Kolter, 2017) proposed to implicitly differentiate the optimal solution with respect to input parameters through the KKT optimality con-

ditions, a set of linear equations that determined the optimal solution. Following this, researchers differentiated through linear programs (Wilder et al., 2019a), submodular optimization problems (Djolonga & Krause, 2017; Wilder et al., 2019a), cone programs (Agrawal et al., 2019a;b), MaxSAT (Wang et al., 2019), Mixed Integer Linear Programming (Ferber et al., 2020; Mandi et al., 2020), Integer Linear Programming (Mandi et al., 2020), dynamic programming solvers Demirovic et al. (2020), blackbox discrete linear optimizers (Pogančić et al., 2019; Rolínek et al., 2020a;b), maximum likelihood estimation (Niepert et al., 2021), kmeans clustering (Wilder et al., 2019b), knapsack (Guler et al., 2022; Demirović et al., 2019), the cross-entropy method (Amos & Yarats, 2020), and SVM training (Lee et al., 2019). Additionally, Wang et al. (2020a) learned to linearly combine LP variables. `SurCo` can use these differentiable surrogates based on the problem domain.

**Task Based Learning**   Task-based learning solves distributions of linear or quadratic optimization problems with the true objective hidden at test time but available for training (Elmachtoub & Grigas, 2022; Donti et al., 2017; El Balghiti et al., 2019; Liu & Grigas, 2021; Hu et al., 2022). (Donti et al., 2021) predicts and corrects solutions for continuous nonlinear optimization. Bayesian optimization (BO) (Shahriari et al., 2016), optimizes blackbox functions by approximating the objective with a learned model that can be optimized over. Recent work optimizes individual instances over discrete spaces like hypercubes (Baptista & Poloczek, 2018), graphs (Deshwal et al., 2021), and MILP (Papalexopoulos et al., 2022). Data reuse from previous runs is proposed to optimize multiple correlated instances (Swersky et al., 2013; Feurer et al., 2018). However, the surrogate Gaussian Process (GP) models are memory and time intensive in high-dimensional settings. Recent work has addressed GP scalability via gradient updates (Ament & Gomes, 2022); however, it is unclear whether GP can scale in conjunction with combinatorial solvers. Machine learning is also used to guide combinatorial algorithms. Several approaches produce combinatorial solutions (Zhang & Dietterich, 1995; Khalil et al., 2017; Kool et al., 2018; Nazari et al., 2018; Zha et al., 2022a;b). Here, approaches are limited to simple feasible regions by iteratively building solutions for problems like routing, assignment, or covering. However, these approaches fail to handle more complex constraints. Other approaches set parameters that improve solver runtime (Khalil et al., 2016; Bengio et al., 2021).

**Learning Latent Space for Optimization**   As we learn latent linear objectives to optimize nonlinear functions, other approaches learn latent embeddings for faster solving. Faloutsos & Lin (1995) proposed FastMap, which learns latent object embeddings for efficient search. Variants of FastMap are used in graph optimization and shortest path (Cohen et al., 2018; Hu et al., 2022; Li et al., 2019). Wang et al. (2020b; 2021a); Yang et al. (2021); Zhao et al. (2022) use monte carlo tree search to perform single and multi-objective blackbox optimization by learning to split the search space.

**Mixed Integer Nonlinear Programming (MINLP)**   `SurCo-zero` falls into the broad family of MINLP solvers, optimizing nonlinear and nonconvex objectives over discrete linear feasible regions. Specialized solvers handle many problem variants in the MINLP space (Burer & Letchford, 2012; Belotti et al., 2013); however, scalabliliy in the nonconvex setting is usually obtained by optimization experts who rely on problem-specific solving techniques such as making piecewise linear approximations, convexifying the objective, or exploiting special structure.

## 6   CONCLUSION

We introduced `SurCo`, a method for learning linear surrogates for combinatorial nonlinear optimization problems. `SurCo` learns linear objective coefficients for a surrogate solver which results in solutions that minimize the nonlinear loss via gradient descent. At its core, `SurCo` differentiates through the surrogate solver which maps the predicted coefficients to a combinatorially feasible solution, combining the flexibility of gradient-based optimization with the structure of combinatorial solvers. We presented three variants of `SurCo`, `SurCo-zero` which optimizes individual instances, `SurCo-prior` which trains a coefficient prediction model offline, and `SurCo-hybrid` which fine-tunes the coefficients predicted by `SurCo-prior` on individual test instances. While `SurCo`'s performance is somewhat limited to binary problems due to the lack of interior integer points, we find that many real-world domains operate on binary decision variables. We evaluated variants of `SurCo` on two domains against the state of the art approaches used in industry, obtaining better solution quality for similar or better runtime in the embedding table sharding domain, and quickly identifying viable photonic devices.

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

## A  PROOFS

**Lemma 3.1** (Sufficient condition of prediction with $\epsilon$-accuracy). *If the dataset $\mathcal{D}_{\mathrm{direct}}$ $(\epsilon/L)$-cover $Y$, then for any $\boldsymbol{y} \in Y$, a 1-nearest-neighbor regressor $\hat{\boldsymbol{\phi}}$ leads to $\|\hat{\boldsymbol{\phi}}(\boldsymbol{y}) - \boldsymbol{\phi}(\boldsymbol{y})\|_2 \leq \epsilon$.*

*Proof.* Since the dataset is a $\epsilon/L$-cover, for any $\boldsymbol{y} \in Y$, there exists at least one $\boldsymbol{y}_i$ so that $\|\boldsymbol{y} - \boldsymbol{y}_i\|_2 \leq \epsilon/L$. Let $\boldsymbol{y}_{\mathrm{nn}}$ be the nearest neighbor of $\boldsymbol{y}$, and we have:

$$\|\boldsymbol{y} - \boldsymbol{y}_{\mathrm{nn}}\|_2 \leq \|\boldsymbol{y} - \boldsymbol{y}_i\|_2 \leq \epsilon/L \tag{7}$$

From the Lipschitz condition and the definition of 1-nearest-neighbor classifier ($\hat{\boldsymbol{\phi}}(\boldsymbol{y}) = \boldsymbol{\phi}(\boldsymbol{y}_{\mathrm{nn}})$), we know that

$$\|\boldsymbol{\phi}(\boldsymbol{y}) - \hat{\boldsymbol{\phi}}(\boldsymbol{y})\|_2 = \|\boldsymbol{\phi}(\boldsymbol{y}) - \boldsymbol{\phi}(\boldsymbol{y}_{\mathrm{nn}})\|_2 \leq L\|\boldsymbol{y} - \boldsymbol{y}_{\mathrm{nn}}\|_2 \leq \epsilon \tag{8}$$

$\square$

**Lemma 3.2** (Lower bound of sample complexity for $\epsilon/L$-cover). *To achieve $\epsilon/L$-cover of $Y$, the size of the training set $N \geq N_0(\epsilon) := \frac{\mathrm{vol}(Y)}{\mathrm{vol}_0} \left(\frac{L}{\epsilon}\right)^d$, where $\mathrm{vol}_0$ is the volume of unit ball in d-dimension.*

*Proof.* We prove by contradiction. If $N < N_0(\epsilon)$, then for each training sample $(\boldsymbol{y}_i, \boldsymbol{\phi}_i)$, we create a ball $B_i := B(\boldsymbol{y}_i, \epsilon/L)$. Since

$$\mathrm{vol}\left(\bigcup_{i=1}^N B_i \cap Y\right) \leq \mathrm{vol}\left(\bigcup_{i=1}^N B_i\right) \leq \sum_{i=1}^N \mathrm{vol}(B_i) = N\mathrm{vol}_0\left(\frac{\epsilon}{L}\right)^d < \mathrm{vol}(Y) \tag{9}$$

Therefore, there exists at least one $\boldsymbol{y} \in Y$ so that $\boldsymbol{y} \notin B_i$ for any $1 \leq i \leq N$. This means that $\boldsymbol{y}$ is not $\epsilon/L$-covered. $\square$

**Theorem 3.1** (A case of infinite Lipschitz constant). *If the minimal distance $d_{\min}$ for different connected components of $\boldsymbol{\phi}(Y)$ is strictly positive, and $\kappa(\boldsymbol{\phi}(Y)) > \kappa(Y)$, then the Lipschitz constant of the mapping $\boldsymbol{\phi}$ is infinite.*

*Proof.* Let $R_1, R_2, \ldots, R_K$ be the $K = \kappa(\boldsymbol{\phi}(Y))$ connected components of $\boldsymbol{\phi}(Y)$, and $Y_1, Y_2, \ldots, Y_J$ be the $J = \kappa(Y)$ connected components of $Y$. From the condition, we know that $\min_{k \neq k'} \mathrm{dist}(R_k, R_{k'}) = d_{\min} > 0$.

We have $R_k \cap R_{k'} = \emptyset$ for $k \neq k'$. Each $R_k$ has a pre-image $S_k := \boldsymbol{\phi}^{-1}(R_k) \subseteq Y$. These pre-images $\{S_k\}_{k=1}^K$ form a partition of $Y$ since

- $S_k \cap S_{k'} = \emptyset$ for $k \neq k'$ since any $\boldsymbol{y} \in Y$ cannot be mapped to more than one connected components;

- $\bigcup_{k=1}^K S_k = \bigcup_{k=1}^K \boldsymbol{\phi}^{-1}(R_k) = \boldsymbol{\phi}^{-1}\left(\bigcup_{k=1}^K R_k\right) = \boldsymbol{\phi}^{-1}(\boldsymbol{\phi}(S)) = S$.

Since $K = \kappa(\boldsymbol{\phi}(Y)) > \kappa(Y)$, by pigeonhole principle, there exists one $Y_j$ that contains at least part of the two pre-images $S_k$ and $S_{k'}$ with $k \neq k'$. This means that

$$S_k \cap Y_j \neq \emptyset, \quad S_{k'} \cap Y_j \neq \emptyset \tag{10}$$

Then we pick $\boldsymbol{y} \in S_k \cap Y_j$ and $\boldsymbol{y}' \in S_{k'} \cap Y_j$. Since $\boldsymbol{y}, \boldsymbol{y}' \in Y_j$ and $Y_j$ is a connected component, there exists a continuous path $\gamma : [0,1] \mapsto Y_j$ so that $\gamma(0) = \boldsymbol{y}$ and $\gamma(1) = \boldsymbol{y}'$. Therefore, we have $\boldsymbol{\phi}(\gamma(0)) \in R_k$ and $\boldsymbol{\phi}(\gamma(1)) \in R_{k'}$. Let $t_0 := \sup\{t : t \in [0,1], \boldsymbol{\phi}(\gamma(t)) \in R_k\}$, then $0 \leq t_0 < 1$. For any sufficiently small $\epsilon > 0$, we have:

- By the definition of $\sup$, we know there exists $t_0 - \epsilon \leq t' \leq t_0$ so that $\boldsymbol{\phi}(\gamma(t')) \in R_k$.

- Picking $t'' = t_0 + \epsilon < 1$, then $\boldsymbol{\phi}(\gamma(t'')) \in R_{k''}$ with some $k'' \neq k$.

On the other hand, by continuity of the curve $\gamma$, there exists a constant $C(t_0)$ so that $\|\gamma(t') - \gamma(t'')\|_2 \leq C(t_0)\|t' - t''\|_2 \leq 2C(t_0)\epsilon$. Then we have

$$L = \max_{\boldsymbol{y},\boldsymbol{y}' \in Y} \frac{\|\boldsymbol{\phi}(\boldsymbol{y}) - \boldsymbol{\phi}(\boldsymbol{y}')\|_2}{\|\boldsymbol{y} - \boldsymbol{y}'\|_2} \geq \frac{\|\boldsymbol{\phi}(\gamma(t')) - \boldsymbol{\phi}(\gamma(t''))\|_2}{\|\gamma(t') - \gamma(t'')\|_2} \geq \frac{d_{\min}}{2C(t_0)\epsilon} \to +\infty \qquad (11)$$

$\square$

## B  EXPERIMENT DETAILS

### B.1  SETUPS

Experiments are performed on a cluster of identical machines, each with 4 Nvidia A100 GPUs and 32 CPU cores, with 1T of RAM and 40GB of GPU memory. Additionally, we perform all operations in Python (Van Rossum & Drake, 2009) using Pytorch (Paszke et al., 2019). For embedding table placement, the nonlinear cost estimator is trained for 200 iterations and the offline-trained models of Dreamshard and `SurCo-prior` are trained against the pretrained cost estimator for 200 iterations. The DLRM Dataset Naumov et al. (2019) is available at `https://github.com/facebookresearch/dlrm_datasets`, and the dreamshard (Zha et al., 2022b) code is available at `https://github.com/daochenzha/dreamshard`. Additional details on dreamshard's model architecture and features can be obtained in the paper and codebase. Training time for the networks used in `SurCo-prior` and `SurCo-hybrid` are on average 8 hours for the inverse photonic design settings and 6, 21, 39, 44, 50, 63 minutes for DLRM 10, 20, 30, 40, 50, 60 settings respectively.

### B.2  NETWORK ARCHITECTURES

#### B.2.1  EMBEDDING TABLE SHARDING

The table features are the same used in Zha et al. (2022b), and sinusoidal positional encoding Vaswani et al. (2017) is used as device features so that the learning model is able to break symmetries between the different tables and effectively group them onto homogeneous devices. The table and device features are concatenated and then fed into Dreamshard's initial fully-connected table encoding module to obtain scalar predictions $\hat{c}_{t,d}$ for each desired objective coefficient. The architecture is trained with the Adam optimizer with learning rate 0.0005.

#### B.2.2  INVERSE PHOTONIC DESIGN

**Network architectures**. The input design specification (a 2D image) is passed through a 3 layer convolutional neural network with ReLU activations and a final layer composed of filtering with the known brush shape. Then a tanh activation is used to obtain surrogate coefficients $\hat{c}$, one component for each binary input variable. The architecture is trained with the Adam optimizer with learning rate 0.001.

This is motivated by previous work (Schubert et al., 2022) that also uses the fixed brush shape filter and tanh operation to transform the latent parameters into a continuous solution that is projected onto the space of physically feasible solutions.

In each setting, optimization is done on a binary grid of different sizes to meet fabrication constraints, namely that a 3 by 3 cross must fit inside each fixed and void location. In the beam splitter the design is an $80 \times 60$ grid, in mode converter it is a $40 \times 40$ grid, in waveguide bend it is a $40 \times 40$ grid, in wavelength division multiplexer it is an $80 \times 80$ grid.

Previous work formulated the projection as finding a discrete solution that minimized the dot product of the input continuous solution and proposed discrete solution. The authors then updated the continuous solution by computing gradients of the loss with respect to the discrete solution and using pass-through gradients to update the continuous solution. By comparison, our approach treats the projection as an optimization problem and updates the objective coefficients so that the resulting projected solution moves in the direction of the desired gradient.

| Task | Randomization |
|------|---------------|
| mode converter | randomize the right and left waveguide width |
| bend setting | randomize the waveguide width and length |
| beam splitter | randomize the waveguide separation, width and length |
| wavelength division multiplexer | randomize the input and output waveguide locations |

Table 1: Task randomization of 4 different tasks in inverse photonic design.

To compute the gradient of this blackbox projection solver, we leverage the approach suggested by Pogančić et al. (2019) which calls the solver twice, once with the original coefficients, and again with coefficients that are perturbed in the direction of the incoming solution gradient as being an "improved solution". The gradient with respect to the input coefficients are then the difference between the "improved solution" and the solution for the current objective coefficients.

## C  Pseudocode

Here is the pseudocode for the different variants of our algorithm. Each of these leverage a differentiable optimization solver to differentiate through the surrogate optimization problem.

---

**Algorithm 1** `SurCo-zero`

---

**Input:** $\Omega, y, f$
1: $c \leftarrow \text{init\_surrogate\_coefs}(y)$
2: **while** not converged **do**
3:      $x \leftarrow \arg\min_{x \in \Omega(y)} c^\top x$
4:      loss $\leftarrow f(x; y)$
5:      $c \leftarrow \text{grad\_update}(c, \nabla_c \text{loss})$
6: **end while**
7: **return** $x$

---

**Algorithm 2** `SurCo-prior` Training

---

**Input:** $\Omega, \mathcal{D}_{\text{train}} = \{y_i\}_{i=1}^N, f$
1: $\theta \leftarrow \text{init\_surrogate\_model}()$
2: **while** not converged **do**
3:      Sample batch $B = \{y_i\}_i^k \sim \mathcal{D}_{\text{train}}$
4:      **for** $y \in B$ **do**
5:          $\hat{c} \leftarrow \hat{c}(y; \theta)$
6:          $x \leftarrow \arg\min_{x \in \Omega(y)} c^\top x$
7:          loss $+= f(x; y)$
8:      **end for**
9:      $\theta \leftarrow \text{grad\_update}(\theta, \nabla_\theta \text{loss})$
10: **end while**

---

**Algorithm 3** `SurCo-prior` Deployment

---

**Input:** $\Omega, \mathcal{D}_{\text{train}} = \{y_i\}_{i=1}^N, f, y_{\text{test}}$
1: $\theta \leftarrow \text{train } \texttt{SurCo-prior}(\Omega, \mathcal{D}_{\text{train}}, f)$
2: $c \leftarrow \hat{c}(y; \theta)$
3: $x \leftarrow \arg\min_{x \in \Omega(y)} c^\top x$
4: **return** $x$

---

**Algorithm 4** `SurCo-hybrid`

---

**Input:** $\Omega, \mathcal{D}_{\text{train}} = \{y_i\}_{i=1}^N, f, y_{\text{test}}$
1: $\theta \leftarrow \text{train } \texttt{SurCo-prior}(\Omega, \mathcal{D}_{\text{train}}, f)$
2: $c \leftarrow \hat{c}(y; \theta)$
3: **while** not converged **do**
4:      $x \leftarrow \arg\min_{x \in \Omega(y)} c^\top x$
5:      loss $\leftarrow f(x; y)$
6:      $c \leftarrow \text{grad\_update}(c, \nabla_c \text{loss})$
7: **end while**
8: **return** $x$

---

## D  Additional Failed Baselines

**SOGA - Single Objective Genetic Algorithm**  Using PyGAD (Gad, 2021), we attempted several approaches for both table sharding and inverse photonics settings. While we were able to obtain feasible table sharding solutions, they underperformed the greedy baseline by $20\%$. Additionally, they were unable to find physically feasible inverse photonics solutions. We varied between random, swap, inversion, and scramble mutations and used all parent selection methods but were unable to find viable solutions.

**DFL - A Derivative-Free Library**   We could not easily integrate DFLGEN (Liuzzi et al., 2015) into our pipelines since it operates in fortran and we needed to specify the feasible region with python in the ceviche challenges. DFLINT works in python but took more than 24 hours to run on individual instances which reached a timeout limit. We found that the much longer runtime made this inapplicable for the domains of interest.

**Nevergrad**   We enforced integrality in Nevergrad (Rapin & Teytaud, 2018) using choice variables which selected between 0 and 1. This approach was unable to find feasible solutions for inverse photonics in less than 10 hours. For table sharding we obtained solutions by using a choice variable for each table, selecting one of the available devices. This approach was not able to outperform the greedy baseline and took longer time so it was strictly dominated by the greedy approach.

