# OpenReview forum: "SurCo: Learning Linear Surrogates for Combinatorial Nonlinear Optimization Problems"
_ICLR.cc/2023/Conference — Submitted to ICLR 2023_

### Official Review · Reviewer_n7J2 · 2022-10-24

**Confidence:** 3
**Correctness:** 3
**Technical Novelty And Significance:** 2
**Empirical Novelty And Significance:** 2
**Recommendation:** 6

**Clarity, Quality, Novelty And Reproducibility:**

The description of the methods is clear. However, the details on the offline dataset generation and training details for SurCO-prior (in turn, SurCO-HYBRID) are not given enough to replicate the experiments. Also, some justification for why linear surrogate learning is better than direct solution learning is not well-conveyed.

**Strength And Weaknesses:**

### Strengths
- The paper presents a way to exploit readily available and efficient MILP solvers to solve even nonlinear problems. Especially, the usage of such solvers handles the feasibility of the solution, i.e. integer constraints, seamlessly and efficiently.
- While its basic for SurCO-zero efficiently utilizes off-the-shelf MILP solvers, its combination with SurCO-prior, SurCO-HYBRID provides a meta-learning approach enabling efficient optimization of the same type of problems by transferring knowledge of similar problems.
- The derivation of SurCO-prior is well-motivated from the regularization perspective.
- The authors rigorously show the difficulty of the widely adapted approach &mdash; using machine learning to learn a mapping from problem instances to solutions.


### Weaknesses
- Limited empirical evaluations
    - Even though the authors admit that there is no performance guarantee, the authors say 'we argue that optimizing Eqn. 2 is better than optimizing the original nonlinear cost'. Considering that the experiments were conducted on two types of problems, this argument is not convincingly supported by extensive empirical analysis or some discussion on the rationale behind it.
- Any discussion on how easier SurCO formulation is compared with direct solution learning?
    - In 3.4, it is shown how difficult it is to learn a mapping from problem instances to an optimum. However, in a sense, SurCO finds an alternative to the optimum solution &mdash; weights of a linear model which can give an optimal solution. Since both have the same dimensionality, on the surface, the reasons for the benefit of SurCO over direction solution learning is not clear. Can you elaborate on this more?
- Reporting more details on the offline data used to fit SurCO-prior.
    - I guess that the number of data, training time, hyperparameters, etc. may affect the performance of SurCO-prior and SurCO-HYBRID significantly. Since the gain of it does not come for free, reporting such detail will be fairer to compare its runtime-related efficiency with baselines.
    - How was the offline data for SurCO-prior in inverse photonic design generated?


### Others
- Offline data for SurCO-prior generation
    - It is interesting that in embedding table sharding, SurCO-prior using offline data generated from DreamShard outperforms DreamShard quite significantly. Maybe some discussion on this highlights further the benefit of the approach.

**Summary Of The Paper:**

The paper propose, SurCO, a combinatorial solver using adaptively tuned linear surrogate models. SurCO utilizes off-the-shelf mixed integer linear program solvers and the differentiability through such solvers. With recent advances in differentiation through optimization layers, SurCO enables end-to-end differentiable training of linear surrogate models. Moreover, the authors propose a meta-learning approach, SurCO-HYBRID, that utilizes the optimization results of previously solved ones for more efficient optimization in new but similar problems. The paper provides a rigorous argument why directly learning a mapping from a problem instance to an optimum is difficult. SurCO is tested on two real-world combinatorial optimization problems.

**Summary Of The Review:**

The paper proposes an interesting way to solve combinatorial problems which can get the best of both, well-developed combinatorial solvers with their long history and data-driven combinatorial solver. The proposed method SurCO-HYBRID is quite appealing even though some intuition or rationale behind it can be discussed better. However, a weak theoretical justification can be made up by more extensive empirical analysis. Weak empirical analysis and not enough details on the experiment need to be improved. If those concerns can be addressed by experiments on other benchmarks or convincing discussions on the benefit of the method, I would increase my score.

---

> ### Author Response · Authors · 2022-11-11
> **Response to n7J2**
>
> Please check the general reply for the rationale of how SurCo can work and how it is better than direct solution learning. Note that even if the dimension is the same, the mapping between $y\rightarrow c$ can be highly nonlinear, which could change the property of the resulting objective substantially.
>
> We try SurCo on two real-world industry-level problems and plan to explore many real-world problems in the future.
>
> We will include additional information regarding empirical evaluation of the training time, hyperparameters. Training data size is given since we are training with the solver in the loop on the number of training instances.
>
> While both SurCo and Dreamshard use offline data to train the value estimator, they optimize the value estimator with different techniques. SurCo learns a linear surrogate, and Dreamshard models it as a sequential decision making problem, and trains it with Reinforcement Learning, which often requires a lot of samples to get it working. In comparison, SurCo is more sample efficient, more stable during training and exhibits relatively smooth convergence.

---

> > ### Comment · Reviewer_n7J2 · 2022-11-29
> > **Response to the rebuttal**
> >
> > Thanks for addressing the questions I have.
> >
> > Even though the authors provide some intuitive answers on why learning through a linear surrogate is better, the theoretical analysis in Subsection 3.4 only shows why directly learning solution is hard and does not show why learning through a linear surrogate is better. I want to ask the authors to state this clearly in the revised version.
> >
> > I think the lack of theory pointed out above is something marginal with respect to the practical value of SurCO. Also, the proposed method is technically sound. So I increase my score for the acceptance. The reason that I cannot increase the score further is that, as I said in the review, the empirical evidence is not extensive enough considering that the value of SurCO is mostly supported by experiments.

---

### Official Review · Reviewer_qvJ5 · 2022-10-25

**Confidence:** 3
**Correctness:** 3
**Technical Novelty And Significance:** 4
**Empirical Novelty And Significance:** 3
**Recommendation:** 8

**Clarity, Quality, Novelty And Reproducibility:**

The paper is clearly written and sufficiently novel. I encourage to provide the source code for helping in reproducibility.

**Strength And Weaknesses:**

- The paper considers an important problem with many real world applications. Experiments demonstrate the efficacy of the proposed approach on challenging domains.

- I found the paper to written quite well with clear description and motivation of the proposed approach. It will be really useful to the reader if a pseudo-code like algorithmic description of the key steps is added in the main paper.

- It is a little suprising to not see any discussion around Bayesian optimization (1) techniques which are quite relevant for this setting. In fact, similar ideas of incorporating combinatorial solvers (not in an end-to-end procedure) have been investigated in the Bayesian optimization literature (Mixed integer program in [2], semi-definite programming in [3], submodular optimization in [4]). Moreover, the setting of SURCO-PRIOR with multiple problem instances can be handled by multi-task or meta Bayesian optimization ([5, 6]). Gaussian processes are the go-to-choice in Bayesian optimization and there is some work on extending GPs to incorporate first-order gradient information as well ([7]). Please consider contextualizing the proposed approach along some of this related work. I feel most of these approaches cannot handle high dimensionality but still it would be nice to discuss the relevance.



References

[1] Shahriari, Bobak, Kevin Swersky, Ziyu Wang, Ryan P. Adams, and Nando De Freitas. "Taking the human out of the loop: A review of Bayesian optimization." Proceedings of the IEEE 104, no. 1 (2015): 148-175.

[2] Papalexopoulos, Theodore P., Christian Tjandraatmadja, Ross Anderson, Juan Pablo Vielma, and David Belanger. "Constrained discrete black-box optimization using mixed-integer programming." In International Conference on Machine Learning, pp. 17295-17322. PMLR, 2022.

[3] Baptista, Ricardo, and Matthias Poloczek. "Bayesian optimization of combinatorial structures." In International Conference on Machine Learning, pp. 462-471. PMLR, 2018.

[4] Deshwal, Aryan, Syrine Belakaria, and Janardhan Rao Doppa. "Mercer features for efficient combinatorial Bayesian optimization." In Proceedings of the AAAI Conference on Artificial Intelligence, vol. 35, no. 8, pp. 7210-7218. 2021.

[5] Swersky, Kevin, Jasper Snoek, and Ryan P. Adams. "Multi-task bayesian optimization." Advances in neural information processing systems 26 (2013).

[6] Feurer, Matthias, Benjamin Letham, and Eytan Bakshy. "Scalable meta-learning for Bayesian optimization." stat 1050, no. 6 (2018).

[7] Ament, Sebastian E., and Carla P. Gomes. "Scalable First-Order Bayesian Optimization via Structured Automatic Differentiation." In International Conference on Machine Learning, pp. 500-516. PMLR, 2022.

**Summary Of The Paper:**

The paper considers the problem of non-linear optimization with combinatorial constraints. A surrogate model based approach is proposed where the key idea is to learn a linear model of the underlying objective which is optimized via combinatorial solvers (for e.g. mixed integer program solvers SCIP) to generate a solution. This training is done in an end-to-end differentiable manner where the gradients are passed through the combinatorial solver generating the solution conditioned on the input weights (denoted as cost in the paper) of the linear model. There are three classes of the proposed approach: (1) SURCO-ZERO, which is applicable to individual instances of a problem, (2) SURCO-prior, which is applicable to a family of problem instances and (3) SURCO-HYBRID, which warm-starts the surrogate weights with that obtained from SURCO-prior. Experiments are performed on two real-world benchmarks: embedding table sharding and inverse photonic design.


**Summary Of The Review:**

Overall, I found the proposed approach interesting and novel while addressing an important and relevant problem for the ICLR community.

---

> ### Author Response · Authors · 2022-11-11
> **Response to qvJ5**
>
> Thank you for your suggestion, we have added pseudocode descriptions of our algorithm variants in the appendix for clarity.
>
> Thank you for pointing out bayesian optimization as another method for optimizing expensive blackbox functions. We have added a section describing the related work in that space. To our knowledge, the algorithms that solve combinatorial settings operate on individual problem instances. Additionally, the meta learning approaches generally leverage an underlying assumption that the different objectives are somewhat correlated such as leveraging results from training models on a small image dataset to generalize to optimizing model hyperparameters for the same image dataset with more samples. However, in our settings the different nonlinear functions are related but not necessarily correlated.

---

### Official Review · Reviewer_r1VR · 2022-11-02

**Confidence:** 4
**Correctness:** 2
**Technical Novelty And Significance:** 2
**Empirical Novelty And Significance:** 3
**Recommendation:** 5

**Clarity, Quality, Novelty And Reproducibility:**

**Clarity & Quality**
The paper is generally well organized and well written. However there are several claims that need to be clarified and better motivated (see questions below).

**Novelty**
The novelty of the paper is the idea of introducing a linear surrogate to replace the variable of the non-linear optimization problem.

**Reproducibility**
Links to the datasets are provided. The precise description of the models hyperparameters is not provided.

**Questions**
1. Sec 3.1: “it also helps escape from local minima, thanks to the embedded search component of existing combinatorial solvers”. To which local minima the paper is referring to here? The MIP solver indeed returns a global minimum for (3). But I don’t see the implication on escaping local minima w.r.t c in Eq (2).
2. Sec 3.2: “the N optimization procedures in the data collection stage are independent of each other, and can lead to excessive number of calls to f that are not helpful. ”. Data collection is solving N SurCo-Zero problems. Why would that lead to an excessive number of f evaluations? Isn't having a labeled training set a requirement of the proposed SurCo-Prior-lambda approach?
3. Sec 3.2: There is a confusion with the c_i: they are first introduced as being the labels in the training set {(y_i, c_i)} then $c_i$’s appear in the variables in Eq (4); then they don’t appear in Eq (5) although it is said “given the training set Dtrain”. Is the training set different at the end of this section?
4. Sec 3.2: “if $\hat{c}$ is a mapping to global optimal solution of c, then it will pull the solutions out of local optima to re-target towards global ones, even when starting from poor initialization, yielding fast convergence and better final solutions for individual optimization instances.” What is meant here by “a mapping to global optimal solution of c”?
5. Sec 3.2: What’s the advantage of “SurCo-prior-λ” (Eq 4) w.r.t. “SurCo-prior” (Eq 5)?
6. Sec 3.2: “but at test time only require the feasible region and not the nonlinear objective.” Since the y is required to predict the c and the objective is defined as a family f(x,y), it looks characterized by y, then I don't understand what is meant by not requiring the objective at test time.
7. Is there any theoretical guarantee or justification on the number of calls needed to optimize f (Eq 1) versus f(g(c)) (ie Eq 2)?
8. Sec 3.4.2: “the mapping y → c(y) can avoid too many connected components in its image c(Y ), by connecting disjoint components of x∗(Y ) together.” What does “too many” means here? Why would this mapping connect disjoint components?
9. Regarding the baselines:
   * Why not using SCIP directly? The paper mentions that scale is a challenge for MINLP solvers but at least for the Embedding Table Sharding problem, it seems that the largest instances have 60x4=240 variables, which should be fine for SCIP?
   * What’s the motivation of using derivative-free methods for optimizing differentiable functions? (Appendix)


**Strength And Weaknesses:**

**Strengths**
1. The paper is well-written
1. It addresses an important and challenging problem: non-linear combinatorial optimization
1. Strong empirical results: clear improvements over the baselines in 2 problems.

**Weaknesses**
1. The paper proposes a surrogate g(c) that is used with the non-linear function f — and not instead of f. Theoretically and intuitively, it is not clear why optimizing f(g(c)) (ie Eq 2) is better than directly optimizing f (Eq 1)
   * I agree with the paper that the proposed optimization (Eq 2) allows to easily handle the linear constraints. But MINLP solvers also generally handle linear constraints (e.g. SCIP).
   * I don’t agree or did not understand the other arguments (see questions 1 to 7)

1. The theoretical analysis (Sec 3.4), that compares “learning solutions” versus the proposed “learning surrogate costs” only holds for a nearest neighbor regressor, which is a very special model and not realistic in practice. Therefore I don’t see how it supports the claims of the paper.
   * For example, learning solutions for CO problems is often formulated as an auto-regressive task (e.g. Pointer Networks by [Vinyals et al 2015] or the Attention Model by [Kool et al 2019])
   * Are there any works that use nearest neighbor regression to predict the solutions of CO problems?

1. Important information is missing in the empirical evaluation:
    * How long did the training take, esp. in terms of number of calls to f?
    * Training datasets are very small: 50 instances for the Embedding Table Sharding problem and 25 instances for the Inverse Photonic Design problem. More information about how training converges and esp. if/how overfitting is avoided would be beneficial.
    * Regarding Inverse Photonic Design, it is not clear what's the objective and what are the constraints.
    * It looks like the objective may be the “design misspecification loss”. Which means that the complex constraints are in fact penalized in the objective. This is a fair strategy when feasibility is challenging but then the argument that the proposed method is more able to handle complex constraints than existing works does not seem fair. [Sec 5: “However, these approaches are unable to handle more complex combinatorial constraints that arise in practice such as those in inverse photonic design”]. Or maybe there are other constraints that are handled — hence the need for a clear formulation of the problem.
    * What’s the size of considered instances of the Inverse Photonic Design?






**Summary Of The Paper:**

The paper presents a learning-based approach to solve combinatorial optimization problems with non-linear objectives and linear constraints. The paper introduces a linear surrogate cost function that can be used by existing solvers, and proposes to learn this surrogate cost in order to efficiently approximate the original problem. This idea is developed in two settings, either for solving individual instances or training a surrogate cost prediction model -- that can possibly be fine-tuned at test time. The approach is evaluated on two industrial problems.

**Summary Of The Review:**

I would vote for borderline reject.

The main contribution of the paper is the introduction of a linear surrogate cost in order to approximate non-linear combinatorial optimization problems. Although it leads to very good experimental results, the main idea is not well-motivated in my opinion, neither theoretically nor intuitively. This makes it hard to see why the proposed approach would work in general, beyond the two presented problems. Some parts of the paper lack clarity and several general claims/arguments need to be clarified or better justified.

---

> ### Author Response · Authors · 2022-11-11
> **Response to r1VR**
>
> # Details of encoding the nonlinear f in the two practical scenarios.
>
> In inverse photonics, the objective is obtained via a physics-based electromagnetic simulation of the photonic device, and gradients are obtained via a simulation-based method. While it may be possible to encode this objective into SCIP via careful modeling, it is nontrivial to our knowledge.
>
> In table sharding, the nonlinear function is a neural network. While recent work in neural network verification has shown that piecewise linear neural networks can be encoded as MILP [Lueg et al 2021], and convex relaxations of nonlinear activations can be used as well [Schweidtmann et al 2019], it is currently nontrivial to encode many practical neural networks such as those with different normalization techniques or where relaxations of nonlinear activations do not hold. Moreover, SCIP’s performance on non-quadratic nonlinear problems is stated to not be as robust as the rest of SCIP (https://www.scipopt.org/doc/html/FAQ.php#minlptypes).
>
> Note that the nearest neighbor (NN) here is used as a vehicle to demonstrate that learning a mapping $y\rightarrow c\rightarrow x^*$ can be statistically easier than learning a direct mapping $y\rightarrow x^*$ from the theoretical point of view. The convenience of using NN is that (1) it is non-parametric (i.e., it can fit to any input-output relationship with sufficient data), and (2) it has clear theoretical properties (e.g., Lemma 3.1-3.2) that supports our claims. In comparison, we have no mathematical tools to analyze pointer networks or attention-based networks directly, even if they have been used in real scenarios.
>
> # Specific Questions
> Q1
> Here the “local optima” refers to the local optima of the original nonlinear problem $f(x)$. Since MILP solver typically has a search component (e.g., branch and bound) that explores different parts of the regions, it can compensate for the localness of gradient-based search and escaping local minima.
>
> Q2
> As mentioned in the general reply, during our training of SurCo-prior-λ (and SurCo-hybrid as well), we don’t need a “labeled” training set, but an unlabeled one \{y_i\}, in which only problem specification $y_i$ is provided.
>
> Q3
> Note that $D_{direct}={(y_i, c_i)}$ is a *hypothetical* training set for direct ML approach, which is never used. The real training set $D_{train}={y_i}$ only consists of problem descriptions.
>
> Q4
> Sorry there is a typo: the sentence should be "if $\hat c$ is a mapping to the global optimal solution of $x$", not $c$.
>
> Q5
> We use SurCo-prior since it has one fewer hyperparameter to tune, while still providing strong empirical performance.
>
> Q6
> In the test time, SurCo-prior only requires the problem description of y, but not the nonlinear objective function f(x;y) itself. This is because after training, SurCo-prior has already established a mapping y→c and simply calling combinatorial solver g(c) we get $x^*$.
>
> Q7
> See the general comments why f(g(c)) is a better way to optimize f over direct optimization of f.
>
> Q8
> Consider the following concrete 2D example. The problem description $y = [cos(\phi), sin(\phi)]$ is a 2D unit vector with the angle $\phi\in[0,\pi/2]$, and the feasible region is convex with 3 vertices being (0, 0), (0, 1), (1, 0). The nonlinear objective is simply $(y^\top x)^2$. The *direct* mapping $y\rightarrow x^*$ maps a continuous region of problem descriptions (i.e., a 1/4 unit circle) into disjoint regions of 2 distinctive points ($x^*$=(0,1) and $x^*$=(1,0)). In contrast, there exists a surrogate cost function c=y, and the mapping $y\rightarrow c$ is identity and continuous.
>
> Q9
> 1. please check our general answer to why not using MINLP.
> 2. the motivation is that these methods are commonly used to solve blackbox optimization problems and we wanted to find reasonable baselines that may solve our application domains. In the inverse photonic design, we use numerical gradient (i.e., finite difference) to compute the “derivative” of the nonlinear loss function, which is a complicated function of the result of an electromagnetic solver.

---

> > ### Comment · Reviewer_r1VR · 2022-12-13
> > **Response to rebuttal**
> >
> > I thank the authors for their response. While most of my questions have been answered, my main concerns about the justification of the approach has not been fully addressed.
> >
> > * The authors reply to my **Q1** was “Here the “local optima” refers to the local optima of the original nonlinear problem . Since MILP solver typically has a search component (e.g., branch and bound) that explores different parts of the regions, it can compensate for the localness of gradient-based search and escaping local minima. “
> >
> >   * For a given $c$, the MIP solver returns $x^*$ the global optimum of (3) but to escape local minima of the original problem as claimed, the cost $c$ would need to be a global optimum as well, which is not the case. This is important because this argument justifies the method intuitively, while I believe being wrong.
> >
> > * Regarding **Q4**, in the revised manuscipt, the authors write: "$\hat{c}$ is a mapping to global optimal solution of $x$ then it will pull the solutions out of local optima to re-target towards global ones, even when starting from poor initialization, yielding fast
> > convergence and better final solutions for individual optimization instances."
> >
> >   * By definition $\hat{c}$ maps $(y, \theta)$ to a predicted surrogate cost $c$. How can it be “a mapping to global optimal solution of $x$”? Again this does not make sense to me, while the paper uses it as an (intuitive) argument for fast convergence and better solutions.
> >
> > * My **Q7** was about theoretical guarantee or justification on the number of calls needed to optimize f (Eq 1) versus f(g(c)) (ie Eq 2).
> >
> >    * Authors point the common reply, where they argue for the benefits of learning: “SurCo-prior and SurCo-hybrid can leverage previous optimization instances… In contrast, a direct optimization of f cannot make use of previous instances and needs to optimize the function from scratch."
> >    * I agree. But this does not answer my question about the number of calls needed to optimize f (Eq 1) versus f(g(c)) (ie Eq 2), which is key since $f$ is supposed to be expensive to evaluate.
> >
> > * Similarly my question about the number of calls to $f$ during training and my concerns about overfitting given the extremely small training sets (of 25 or 50 instances) have not been addressed.
> >
> > After reading the new additions in the revised version, the other reviews and authors responses, I would still keep my score with the same arguments as in the summary of my review above.

---

### Author Response · Authors · 2022-11-11
**Common Answers**

We thank the reviewers for their insightful comments. We are glad to see reviewers think the paper addresses an important real-world problems (i.e., the nonlinear optimization with combinatorial constraints) [r1VR, qvJ5, n7J2] and obtains strong empirical results [r1VR, qvJ5] in real-world applications.

Reviewers have concerns on the following and we will address below:

# Why SurCo is better than directly optimizing the original cost [r1VR]?

Compared to directly optimizing the original nonlinear loss f, SurCo-prior and SurCo-hybrid can leverage previous optimization instances (parameterized by $y_i$) to gain knowledge and optimize similar instances in a more efficient manner. In contrast, a direct optimization of f cannot make use of previous instances and needs to optimize the function from scratch.

# Why SurCo is better than learning a direct mapping from problem description $y$ to optimal solution $x^*$ [n7J2] ?

While we could choose to learn a direct mapping (e.g., previous approaches) $y\rightarrow x^*$ (i.e., from the problem description y to optimal solution $x^*$), we show in Sec. 3.4 that a direct mapping may have an infinitely large Lipschitz constant (Theorem 3.1), due to the fact that the optimal solution can jump from one vertex to the other in the presence of combinatorial constraints. Therefore, the number of samples required to learn this mapping using 1-nearest-neighbor is also infinite (Lemma 3.2).

SurCo circumvents this difficulty by taking a detour (learning $y\rightarrow c\rightarrow x^*$) rather than going direct ($y\rightarrow x^*$). Intuitively, even if $y\rightarrow x^*$ can be discontinuous, $y\rightarrow c$ can still be continuous since the combinatorial solver $c\rightarrow x^*$ takes away the discontinuity part. Intuitively, this can be regarded as "lifting" the optimization problem via a nonlinear mapping so that the solution can be obtained via a linear solver, analogous to how kernel SVM works (but the mapping now becomes learnable). Note that this is not a linearization of the original nonlinear problem.

While reviewer n7J2 may have the concerns that c and $x^*$ are of the same dimension, this doesn’t matter since the mapping $c\rightarrow x^*$ is nonlinear and can potentially decouple the complicated nonlinear problems.


# Why not use existing nonlinear solvers (e.g., MINLP) for the two problems [r1VR] ?

Fundamentally, for existing solvers like MINLP, the nonlinear functions and/or the constraints should be written as an analytical mathematical form. E.g., SCIP can only handle addition, subtraction, multiplication, division, exponentiation, and logarithm but not trigonometric functions. This imposes fundamental constraints on the type of nonlinear function f that can be optimized.

However, in many other real-world settings, the objective function f is often a blackbox that outputs a function value (and its gradient), as is the case with our inverse photonic design setting. Encoding f in an analytic form is highly nontrivial or simply impossible, let alone leveraging MINLP. In contrast, SurCo can handle such situations with ease, as one of the key features of our approach.

Empirically, we tried MINLP solvers on the real-world cases but they failed to yield nontrivial solutions.


# Details of experiments. [r1VR, n7J2]

Inverse photonic design has been further specified in the appendix.
The constraints are that both the filled and void pixels in the device must be a union of a 3 x 3 cross-shaped brush to ensure that the brush could “fit” in any filled or void space.

The objective is computed by performing electromagnetic (EM) simulation (i.e., Maxwell’s equation) based on the design, to obtain the EM fields (i.e., intensity of electric and magnetic fields) at all pixels. The difference between the computed and desired strength are computed per-pixel. The overall loss is the euclidean norm of these distances. Once the distances are all 0, then the design meets the specification.

# Data generation

For both problems, we prepare an offline *unlabeled* training set {$y_i$} by random sampling problem specification $y_i$ from a pre-defined distribution. For the evaluation set, from the same distribution we sample specifications that are different from training. Note that in the dataset we don’t need to prepare optimal solution $x^*_i$, since in surco-prior and surco-hybrid, the learning procedure naturally involves finding their optimal solutions (Eqn. 4 and Eqn. 5)

Note that in our theoretical analysis involving 1-nearest-neighbor, we assume a hypothetical labeled dataset $\{(y_i, \phi_i)}$ with both the problem specification $y_i$ and the “target” $\phi_i$ to be regressed (e.g., in Sec. 3.4). However, empirically, SurCo doesn’t require the target in the dataset. We apologize for any confusion caused by the notations inconsistency.

# Other details
We have provided a revised version with the training time and hyperparameters for surco-prior.

---

### Decision · Program_Chairs · 2023-01-20

**Decision:**

Reject

**Justification For Why Not Higher Score:**

The paper proposes a surrogate that is used along with the non-linear function f.
Theoretically and intuitively, it is not clear why optimizing their proposed formulation is better than directly optimizing the original problem.
The theoretical analysis (Sec 3.4), that compares “learning solutions” versus the proposed “learning surrogate costs” only holds for a nearest neighbor regressor (which is a very special model and not realistic in practice).
Important information is missing in the empirical evaluation:
A discussion on how easier SurCO formulation is compared with direct solution learning could be refined.

**Justification For Why Not Lower Score:**

The paper is well-written and addresses an important and challenging problem: non-linear combinatorial optimization
Strong empirical results were proposed, with some improvements over the baselines in two problems.


**Metareview: Summary, Strengths And Weaknesses:**

The paper presents a learning-based approach to solve combinatorial optimization problems with non-linear objectives and linear constraints. It introduces a linear surrogate cost function that can be used by existing solvers, and proposes to learn this surrogate cost to efficiently approximate the original problem.
This idea is developed in two settings, either for solving individual instances or training a surrogate cost prediction model -- t
Training is done in an end-to-end differentiable manner where the gradients are passed through the combinatorial solver generating the solution conditioned on the input weights. The approach is evaluated on two industrial problems.


**Summary Of Ac-Reviewer Meeting:**

No meetings, most reviewers were not active during the discussions phase.